



# Technical Note: Effect of varying the λ = 185 and 254 nm photon flux ratio on radical generation in oxidation flow reactors

Jake P. Rowe[1,+,*], Andrew T. Lambe[2,*], and William H. Brune[1]

[1]Department of Meteorology and Atmospheric Science, Pennsylvania State University, University Park, PA, USA
[2]Center for Aerosol and Cloud Chemistry, Aerodyne Research Inc., Billerica, MA, USA
[+]Now at: Department of Chemistry, University of Colorado, Boulder, CO, USA
[*]These authors contributed equally to this work.

**Correspondence:** Andrew T. Lambe (lambe@aerodyne.com)

**Abstract.**

Oxidation flow reactors (OFRs) complement environmental smog chambers as a portable, low-cost technique for exposing atmospheric compounds to oxidants such as ozone ($O_3$) and hydroxyl (OH) radicals. OH is most commonly generated in OFRs via photolysis of externally added $O_3$ at $\lambda$=254 nm (OFR254), or combined photolysis of $O_2$ and $H_2O$ at $\lambda$ = 185 nm

plus photolysis of $O_3$ at $\lambda$=254 nm (OFR185) using low-pressure mercury (Hg) lamps. Whereas OFR254 radical generation is influenced by [$O_3$], [$H_2O$], and photon flux at $\lambda$ = 254 nm ($I_{254}$), OFR185 radical generation is influenced by [$O_2$], [$H_2O$], $I_{185}$, and $I_{254}$. Because the ratio of photon fluxes, $I_{185}$:$I_{254}$, is OFR-specific, OFR185 performance varies between different systems even when constant $H_2O$ and $I_{254}$ are maintained. Thus, calibrations and models developed for one OFR185 system may not be applicable to another. To investigate these issues, we conducted a series of experiments in which $I_{185}$:$I_{254}$ emitted

by Hg lamps installed in an OFR was systematically varied by fusing multiple segments of lamp quartz together that either transmitted or blocked $\lambda$ = 185 nm radiation. Integrated OH exposure ($OH_{exp}$) values achieved for each lamp type were obtained using the tracer decay method as a function of UV intensity, humidity, residence time, and external OH reactivity ($OHR_{ext}$). Following previous related studies, a photochemical box model was used to develop a generalized $OH_{exp}$ estimation equation as a function of [$H_2O$], [$O_3$] and $OHR_{ext}$ that is applicable for $I_{185}$:$I_{254} \approx 0.001$ to $0.1$.

## 1 Introduction

Hydroxyl (OH) radicals govern the concentrations of most atmospheric organic compounds, including those that lead to secondary organic aerosol (SOA) formation. For decades, environmental chambers and oxidation flow reactors (OFRs) have been used to simulate atmospheric aging processes through the controlled exposure of trace gases and aerosols to OH radicals. Environmental chamber studies are typically conducted over experimental timescales and equivalent atmospheric exposure times

of hours up to 1 or 2 days. OFRs with residence times on the order of minutes achieve multiple days of equivalent atmospheric OH exposure ($OH_{exp}$), typically through the reactions





$$O_3 + h\nu_{254} \quad \rightarrow \quad O_2 + O(^1D) \tag{R1}$$

$$O(^1D) + H_2O \quad \rightarrow \quad 2OH \tag{R2}$$

$$O(^1D) + N_2O \quad \rightarrow \quad 2NO \tag{R3}$$

This method is referred to as OFR254, and relies on addition of externally generated $O_3$ at the OFR inlet. In some cases, OFRs have additionally employed the secondary $\lambda = 185$ nm emission line present in low-pressure mercury (Hg) lamps to generate radicals from the following reactions in addition to those listed above that are employed in OFR254:

$$H_2O + h\nu_{185} \quad \rightarrow \quad H + OH \tag{R4}$$

$$H + O_2 \quad \rightarrow \quad HO_2 \tag{R5}$$

$$O_2 + h\nu_{185} \quad \rightarrow \quad 2O(^3P) \tag{R6}$$

$$O(^3P) + O_2 \quad \rightarrow \quad O_3 \tag{R7}$$

$$N_2O + h\nu_{185} \quad \rightarrow \quad N_2 + O(^1D) \tag{R8}$$

This method is referred to as OFR185. Recent modeling studies suggest that OFR185 is less affected by experimental arti-facts than OFR254 such as SOA photolysis and unwanted reactions with non-OH oxidants (Peng et al., 2016, 2018, 2019).

Additionally, OFR185 is often more practical than OFR254 to apply in field studies because compressed air or $O_2$ is not re-quired for external $O_3$ generation. However, because the $\lambda = 185$ nm photon flux ($I_{185}$) is influenced by OFR-specific design considerations that are mainly related to the Hg lamps being used, concentrations of $O_3$, $HO_x = OH + HO_2$ and $NO_x = NO$ + $NO_2$ generated using OFR185 are potentially variable between different systems even when constant $H_2O$, $N_2O$ and $I_{254}$ are established. Thus, calibrations and models developed for one OFR185 system may not be applicable to another, making it

more difficult to evaluate results or plan experiments. To investigate these issues, we designed a series of experiments in which $I_{185}:I_{254}$ was systematically varied over a wide range using multiple novel Hg lamp configurations. Integrated $OH_{exp}$ values were obtained as a function of OFR185 conditions, and a photochemical box model was used to develop a system of $OH_{exp}$ estimation equations that are applicable to OFR185 systems with $I_{185}:I_{254} \approx 0.001$ to 0.1.

## 2 Experimental

Experiments were conducted using an Aerodyne Potential Aerosol Mass (PAM) OFR, which is a horizontal aluminum cylin-drical chamber (46 cm long × 22 cm ID) operated in continuous flow mode (Lambe et al., 2011). A simplified schematic is shown in Figure S1. The $H_2O$ mixing ratio in the OFR was controlled by passing the carrier gas through a Nafion humidifier (Perma Pure LLC) or heated recirculating water bath (Neslab Instruments, Inc.) and then diluting with different levels of dry carrier gas at the OFR inlet. A photodetector (TOCON-C6, sglux Gmbh) and a relative humidity and temperature (RH/T)

sensor were mounted in the exit flange of the OFR. Across all experiments, [$H_2O$] ranged from 0.03% (1% RH at 25.3 °C) to





3.9% (88% RH at 30.9 °C). The $O_3$ mixing ratio at the exit of the OFR was measured with a UV ozone analyzer (106-M, 2B Technologies).

## 2.1 HO$_x$ generation

HO$_x$ was produced via reactions R1-R2 and R4-R7. Photolysis of $H_2O$, $O_2$ and $O_3$ in the OFR was achieved using two low-pressure Hg fluorescent lamps (Light Sources, Inc.) that were isolated from the sample flow using type 214 quartz sleeves. Nitrogen purge gas was flowed over the lamps to prevent $O_3$ buildup between the lamps and sleeves. A fluorescent dimming ballast was used to regulate current applied to the lamps. The dimming voltage applied to the ballast ranged from 0.8 to 10 VDC.

Figure 1 shows the Hg fluorescent lamp configurations that were used in this study. Lamp type A is a standard ozone-producing low-pressure Hg germicidal fluorescent lamp (GPH436T5VH/4P, Light Sources Inc.) in which quartz that transmits $\lambda$ = 185 and 254 nm radiation is present along the entire 356 mm arc length. The relative transmissivity of $\lambda$ = 185 nm radiation ($T_{185}$) in lamp type A is thus equal to 1. Lamp type B is equivalent to lamp type A with added segments of opaque heat shrink tubing applied to approximately 86% of the arc length ($T_{185} \approx 0.14$; see also Fig. S2) to reduce $I_{185}$ and $I_{254}$ to levels below what is achievable using the ballast dimming voltage. A different type of quartz is available that blocks $\lambda$ = 185 nm and transmits $\lambda$ = 254 nm radiation ($T_{185}$ = 0). Lamp types C, D, E (GPH436T5L/VH/4P 90/10, GPH436T5L/VH/4P 96/4, GPH436T5L/VH/4P 98.5/1.5; Light Sources, Inc.) fused one segment each of quartz with $T_{185}$ = 0 and $T_{185}$ = 1 to provide reduced $I_{185}$ relative to lamp type A while maintaining constant $I_{254}$. Finally, to evaluate the effect of lamp design at fixed $T_{185}$ and $I_{254}$, lamp types F and G contain the same ratios of $T_{185}$ = 0 and $T_{185}$ = 1 quartz as Types C and D, but with 5 and 13 total segments instead of 2 segments. These different designs isolate the effect of discretized $\lambda$ = 185 nm irradiation across the entire arc length of the lamp versus having all $\lambda$ = 185 nm radiation near the entrance of the OFR.

## 2.2 OH$_{exp}$ characterization studies

OH$_{exp}$, defined here as the product of the average OH concentration and the mean OFR residence time ($\tau_{OFR}$), was characterized by measuring the decay of carbon monoxide (CO) and/or sulfur dioxide (SO$_2$) tracers using Thermo 48i and 43i CO and SO$_2$ analyzers (e.g. Lambe et al. (2011)). Tracer mixing ratios entering the reactor were 6-9 ppmv for CO and 288-629 ppbv for SO$_2$, each diluted from separate gas mixtures of 0.5% CO or SO$_2$ in $N_2$ (Praxair). The corresponding total external OH reactivity (OHR$_{ext}$), which is the summed product of each tracer mixing ratio and its bimolecular OH rate coefficient, ranged from approximately 9 to 64 s$^{-1}$. Tracer concentrations were allowed to stabilize before initiating OH$_{exp}$ measurements, during which steady-state levels of CO and/or SO$_2$ were obtained with the lamps turned off. Then, the lamps were turned on, and tracer concentrations were allowed to equilibrate before being measured at illuminated steady-state conditions.

In most experiments, the mean residence time was $\tau_{OFR}$ = 124 s, which was calculated from the ratio of the internal OFR volume ($\approx$ 13 L) and the total sample and makeup flow rate through the OFR (6.4 L min$^{-1}$). This calculation implicitly assumes plug flow conditions, with associated uncertainty of approximately 10% compared to an explicit residence time distribution measurement (Li et al., 2015b). To characterize the uncertainty in the plug flow approximation in our work, in a subset





of experiments, $\tau_{OFR} \approx 63$ and 251 s were achieved by systematically changing the sample flow rate to 12.5 and 3.1 L min$^{-1}$.

At $\tau_{OFR}$ = 63, 124, and 251 s, we measured integrated OH exposure (OH$_{exp}$) values of $3.3 \times 10^{11}$, $7.8 \times 10^{11}$, and $2.0 \times 10^{12}$ molecule cm$^{-3}$ s, respectively, using the tracer decay method (Sec. 2.2) with the OFR operated at the same humidity and lamp intensity. Thus, perturbing the "plug flow" $\tau_{OFR}$ = 124 s by a factor of 2 in either direction changed OH$_{exp}$ by factors of 2.36 and 2.56. Based on these results, an upper-limit estimated uncertainty in $\tau_{OFR}$ and corresponding OH$_{exp}$ is approximately 30%.

## 2.3 Photochemical model

We used a photochemical model implemented in MATLAB and Igor Pro to calculate concentrations of radical/oxidant species produced in the reactor (Li et al., 2015b). The KinSim chemical kinetic solver was used to compile the version of the model that was implemented in Igor Pro (Peng and Jimenez, 2019). Model input parameters are shown in Table 1, and reactions and associated kinetic rate coefficients that were included in the model are summarized in Table S1 (Peng and Jimenez, 2020).

For cases where [H$_2$O] $\leq$ 0.1% and the RH was comparable to the accuracy of the measurement ($\pm$ 2% RH), [H$_2$O] input to the model was varied between 0.01% and 0.1% to generate better model:measurement OH$_{exp}$ agreement. $I_{254}$ and $I_{185}$ values input to the model were adjusted to match the measured OH$_{exp}$ values as best as possible within the following constraints:

1. $I_{254,max} = (3.5 \pm 0.7) \times 10^{15}$ photons cm$^{-2}$ s$^{-1}$ for two lamps operated at maximum output (Lambe et al., 2019).

2. At reduced lamp output, $I_{254}$ was calculated by multiplying $I_{254,max}$ by the ratio of photodetector-measured irradiance
values measured at maximum and reduced lamp output at $\lambda$ = 254 nm.

3. $I_{185,max}$:$I_{254,max} \lesssim$ 0.10 for lamp types A and B only (Spicer, 2013).

4. $I_{185,max}$:$I_{254,max} \lesssim$ 0.01, 0.004, and 0.0015 for lamp types C and F, D and G, and E, respectively.

5. At reduced lamp output, I$_{185}$ was calculated by multiplying $I_{185,max}$ by the ratio of O$_3$ mixing ratios measured at maximum and reduced lamp output.

Within these constraints, the mean ($\pm$ 1$\sigma$) ratios of modeled:measured CO and SO$_2$ concentrations remaining at the exit of the OFR were 1.02$\pm$0.06 and 0.97$\pm$0.17, respectively.

## 3 Results and Discussion

### 3.1 Influence of $I_{185}$ on [O$_3$] and OH$_{exp}$

Figure 2 shows [O$_3$] measured at the exit of the OFR as a function of T$_{185}$ with each lamp type operated at maximum UV
output. Binned data are shown for conditions where [H$_2$O] = 0.15$\pm$0.11%, 0.98$\pm$0.08%, 1.74$\pm$0.23%, and 3.42$\pm$0.30%. At fixed [H$_2$O], [O$_3$] increased as a function of T$_{185}$. For example, [O$_3$] increased from 17.8 to 155 ppmv at [H$_2$O] = 0.15% and from 4.5 to 56 ppmv at [H$_2$O] = 1.74% as T$_{185}$ increased from 0.1 to 1. At fixed T$_{185}$ and $I_{254}$, [O$_3$] decreased with increasing





[H$_2$O] due to faster O($^1$D) + H$_2$O reaction rate following O$_3$ photolysis at $\lambda$ = 254 nm. Consequently, as [H$_2$O] increased from 0.15 to 3.42%, [O$_3$] decreased by a factor of 4-5 for lamp types A and C-G, whereas [O$_3$] decreased by a factor of 2

for lamp type B because of its reduced I$_{254}$ (Fig. 1). At [H$_2$O] = 1.74% and T$_{185}$ = 0.04 and 0.1, Figure 2, shows that [O$_3$] generated using lamp types D and G was approximately 1.7 and 1.8 ppmv; here, lamp type D had one 15 mm quartz segments with T$_{185}$ = 1, whereas lamp type G had three 5 mm quartz segments with T$_{185}$ = 1. At the same OFR conditions, [O$_3$] generated using lamp types C and F was 4.5 and 2.7 ppmv; these lamps had one 35 mm and seven 5 mm quartz segments with T$_{185}$ = 1. Despite the discrepancy in measured [O$_3$], corresponding OH$_{exp}$ obtained with lamp types C and F were 2.5×10$^{12}$

and 2.8×10$^{12}$ molecules cm$^{-3}$ s respectively. Thus, the worse agreement in [O$_3$] measured between lamp types C and F may be associated specifically with O$_3$ measurements from these experiments. We hypothesize that the OFR-volume-averaged I$_{185}$ is sufficient to describe associated HO$_x$ production for these cases.

Figure 3 plots OH$_{exp}$ as a function of T$_{185}$ at [H$_2$O]= 1.90±0.26%. The corresponding equivalent photochemical age shown on the right y-axis assumes a 24-hour average OH concentration of 1.5×10$^6$ molec cm$^{-3}$ (Mao et al., 2009). Results obtained

with lamp types D & G and C & F were averaged together at T$_{185}$ = 0.04 and 0.1 respectively due to their similar OH$_{exp}$ values. Over the range of T$_{185}$ shown in Fig. 3, excluding lamp type B, OH$_{exp}$ increased by approximately a factor of 5 at I$_{254}$ = (3.7±0.6)×10$^{15}$ photons cm$^{-2}$ s$^{-1}$) and a factor of 17 at I$_{254}$ = (2.1±0.3)×10$^{14}$ photons cm$^{-2}$ s$^{-1}$. Maximum OH$_{exp}$ also decreased by about a factor of 5 between lamp types A and B due to reduction in both I$_{254}$ and I$_{185}$ (not shown in Fig. 3). Similar trends were observed for OH$_{exp}$ measurements at [H$_2$O] = 0.93±0.06% and 3.42±0.30%, but at [H$_2$O] = 0.09±0.07%,

the sensitivity of OH$_{exp}$ to T$_{185}$ was weaker due to suppressed OH production at lower humidity.

### 3.2    $I_{185}$:$I_{254}$ determination and derivation of OH$_{exp}$ estimation equations

Figure 4 plots $I_{185}$ as a function of $I_{254}$ for the Hg lamps used in this study and a different model of Hg lamps used in an earlier-generation PAM OFR (Li et al., 2015a). As with OH$_{exp}$ values shown in Fig. 3, $I_{185}$ and $I_{254}$ values obtained with lamp types D & G and C & F were combined together into T$_{185}$ = 0.04 and 0.1 symbols following our hypothesis that the

OFR-volume-averaged $I_{185}$ was sufficient to describe HO$_x$ production. Linear fits applied to the data shown in Fig. 4 were used to calculate average $I_{185}$: $I_{254}$ values for lamp types A & B, C & F, D & G, and E. Lamp types A and B (red symbols) had the highest $I_{185}$: $I_{254}$ = 0.0664, whereas lamp type E had the lowest $I_{185}$: $I_{254}$ = 0.00167. $I_{185}$: $I_{254}$ = 0.00561 for lamp types C and F (blue symbols) fell within the envelope of $I_{185}$: $I_{254}$ = 0.004 to 0.012 characterized by Li et al. (2015a), with a lower apparent sensitivity of $I_{185}$: $I_{254}$ to lamp power. This is presumably due to differences in the specific Hg lamps and/or

method of dimming used in the two studies.

Previous studies reported empirical OH exposure algebraic estimation equations for use with OFRs (Li et al., 2015a; Peng et al., 2015, 2018; Lambe et al., 2019). These equations parameterize OH$_{exp}$ as a function of readily-measured experimental parameters, therefore providing a simpler alternative than detailed photohemical models for experimental planning and analysis. Here, we expand on those studies by deriving OH$_{exp}$ estimation equations for the lamp types that were used in this study.



We adapted the estimation equation format introduced by Li et al. (2015a):

$$\log[\mathrm{OH_{exp}}] = \left(\mathrm{a} + (\mathrm{b} + \mathrm{c} \times \mathrm{OHR_{ext}^d} + \mathrm{e} \times \log[\mathrm{O_3} \times \mathrm{OHR_{ext}^f}]) \times \log[\mathrm{O_3}] + \log[\mathrm{H_2O}]\right) + \log\left(\frac{\tau}{124}\right) \tag{1}$$

Equation 1 was fit to data obtained from the base case of the model, with CO reacting with OH as a surrogate of $\mathrm{OHR_{ext}}$, over the following OFR185 phase space: $T = 25°C$, $\tau = 124$ s, $\mathrm{OHR_{ext}} = 0.77$ to 232 s$^{-1}$, $[\mathrm{H_2O}] = 0.1$ to 3%, $I_{254} = 10^{13}$ to $10^{16}$ photons cm$^{-2}$ s$^{-1}$, and $I_{185}{:}I_{254} = 0.00167, 0.00242, 0.00595,$ and $0.0664$. For each $I_{185}{:}I_{254}$ value, we explored 10, 15, and

150 20 logarithmically evenly distributed values in the ranges of $\mathrm{OHR_{ext}}$, $[\mathrm{H_2O}]$, and $I_{254}$, respectively. Figure 5 compares $\mathrm{OH_{exp}}$ estimated from Eq. 1 and calculated from the model for the $I_{185}{:}I_{254} = 0.0664$ case. Almost all of the equation-estimated and model $\mathrm{OH_{exp}}$ values agreed within a factor or 2 or better. The absolute value of the relative deviations increased above $[\mathrm{H_2O}] \approx 0.5\%$ and was largest at $[\mathrm{H_2O}] = 3\%$; the mean absolute value of the relative deviations was 28%. Analogous plots for $I_{185}{:}I_{254} = 0.00167, 0.00242,$ and $0.00595$ cases are shown in Fig. S3. For these other cases, the mean absolute values of the

155 relative deviations were 20%, 17% and 16%, respectively. Eq. 1 coefficients for lamps with the $I_{185}{:}I_{254}$ values reported here are presented in Table 2.

To generalize the results shown in Figs. 5 and S3 to OFR185 systems with other $I_{185}{:}I_{254}$ values, Figure 6 plots fit coefficients $a$ - $f$ as a function of $I_{185}{:}I_{254}$. Each of these coefficients changes monotonically as a function of $I_{185}{:}I_{254}$, enabling the usage of simple exponential regression functions to parameterize the $a$ - $f$ values as a continuous function of $I_{185}{:}I_{254}$.

Exponential function coefficients for the regression curves shown in Fig. 6 are presented in Table 3. Figure 7 compares the equation-estimated $\mathrm{OH_{exp}}$ (obtained using Eq. 1 with Table 3 fit coefficients) and the measured $\mathrm{OH_{exp}}$ obtained using the tracer decay method. The mean ($\pm 1\sigma$) ratios of equation-estimated and measured $\mathrm{OH_{exp}}$ values were $0.94\pm0.55$, $1.13\pm0.48$, $1.03\pm0.37$, and $1.32\pm0.71$ for $I_{185}{:}I_{254} = 0.00167, 0.00242, 0.00595,$ and $0.0664$.

### 3.3 Influence of $I_{185}$ on HO$_2$, NO$_x$, and UV photolysis of aromatic volatile organic compounds

In addition to [OH], [HO$_2$] and [NO$_x$] (with N$_2$O present) also increase with increasing $I_{185}$ (Fig. S4). To isolate the effect of $I_{185}$ on related OFR photochemistry at fixed $\mathrm{OH_{exp}}$, we investigated two OFR185 cases using $(I_{185}, I_{254}) = (3.33\times10^{12}, 1.96\times10^{15})$ and $(6.65\times10^{12}, 1.01\times10^{14})$ photons cm$^{-2}$ s$^{-1}$ that each generate a model-calculated $\mathrm{OH_{exp}} = 5.0\times10^{11}$ molecules cm$^{-3}$ s at base case conditions of $[\mathrm{H_2O}] = 2\%$, $\tau = 124$ s, and $\mathrm{OHR_{ext}} = 30$ s$^{-1}$. These cases were designated as "low" and "high" $I_{185}{:}I_{254}$ cases. Thus, increasing $I_{185}$ by a factor of 2 enabled lowering $I_{254}$ by a factor of 20 to achieve equivalent

$\mathrm{OH_{exp}}$.

First, we investigated the resilience of each OFR185 case to OH suppression via $\mathrm{OHR_{ext}}$. As $\mathrm{OHR_{ext}}$ was increased from 30 to 300 s$^{-1}$, $\mathrm{OH_{exp}}$ decreased from $5.0\times10^{11}$ to $7.9\times10^{10}$ (low $I_{185}{:}I_{254}$) and $9.0\times10^{10}$ (high $I_{185}{:}I_{254}$) molecules cm$^{-3}$ s$^1$. Thus, increasing $I_{185}$ decreased OH suppression by 15%, primarily due to 30% higher [HO$_2$] in the high $I_{185}{:}I_{254}$ case that increased the OH + HO$_2$ reaction rate and partially buffered the system against increasing $\mathrm{OHR_{ext}}$. Second, we compared the

175 ability of each OFR185 case to generate high-NO conditions in the presence of added [N$_2$O]. For example, at [N$_2$O] = 2.7%, NO:HO$_2$ = 1 and 0.4 at low and high $I_{185}{:}I_{254}$. While increasing [N$_2$O] from 2.7% to 4.0% achieved NO:HO$_2$ = 1 at high $I_{185}{:}I_{254}$, [NO$_2$] also increased from 50 to 100 ppbv. At higher UV intensity, a similar increase in [N$_2$O] could generate [NO$_2$]





> 1 ppm and promote artificially fast $RO_2 + NO_2$ reactions compared to atmospheric conditions (Peng and Jimenez, 2017).
Third, we compared relative timescales for OH oxidation and photolysis of representative aromatic volatile organic compounds

(VOCs) that absorb $\lambda$ = 185 and 254 nm radiation. The fractional VOC loss due to photolysis, $F_{photolysis}$, was calculated using
Equation 2:

$$F_{photolysis} = \frac{\sigma_{185}I_{185}\phi_{185} + \sigma_{254}I_{254}\phi_{254}}{\sigma_{185}I_{185} \times \phi_{185} + \sigma_{254}I_{254}\phi_{254} + k_{OH}[OH]} \tag{2}$$

Where $\sigma_{185}$ and $\sigma_{254}$ are the VOC absorption cross sections at $\lambda$ = 185 and 254 nm, $\phi_{185}$ and $\phi_{254}$ are the VOC photolysis
quantum yields, and $k_{OH}$ is the bimolecular reaction rate coefficient with OH. Assuming upper limit $\phi_{185}$ = 1 and $\phi_{254}$

=1 values, $F_{photolysis,benzene} \leq 0.26$ and 0.07 at low and high $I_{185}:I_{254}$ (Atkinson, 1986; Dawes et al., 2017). Similarly,
$F_{photolysis,toluene} \leq 0.07$ and 0.04 at low and high $I_{185}:I_{254}$ (Atkinson, 1986; Serralheiro et al., 2015).

## 4  Conclusions

OFR185 is emerging as one of the most commonly used OFR methods by enabling efficient $HO_x$ and $NO_x$ generation over a
range of oxidative aging timescales that are relevant to atmospheric processes. Important OFR185 parameters are $I_{185}$, $I_{254}$,

$[H_2O]$, $[N_2O]$ (if $NO_x$ generation is required), $OHR_{ext}$, and $\tau_{OFR}$. However, $I_{185}:I_{254}$ is specific to the Hg lamp and/or OFR,
as are associated calibration and estimation equations. To develop a general framework within which to evaluate and compare
different OFR185 systems, we characterized $OH_{exp}$ as a function of $I_{185}$, $I_{254}$, $OHR_{ext}$, and $[H_2O]$ values, in the process using
several novel low-pressure Hg lamp configurations to extend the range of achievable $I_{185}:I_{254}$. $OH_{exp}$ estimation equations
were developed for the Hg lamp types that were used, and corresponding estimation equation fit coefficients were parameterized

as a function of $I_{185}:I_{254}$ to enable interpolation to other OFR185 systems that were not studied here. $OHR_{int}$, $HO_2:OH$, and
$F_{photolysis}$ were improved at higher $I_{185}:I_{254}$, whereas $NO:HO_2$ and $NO:NO_2$ were improved at lower $I_{185}:I_{254}$. Overall, our
results suggest that optimal OFR185 performance is achieved by (1) maximizing $I_{185}:I_{254}$ (2) reducing $OH_{exp}$ (if needed)
through simultaneous reduction in $I_{185}$ and $I_{254}$ via electronically or mechanically dimming the lamp output (Fig. S2) (3)
increasing $[N_2O]$ to offset higher $[HO_2]$ if high-NO conditions are required, provided that $[NO_2]$ does not exceed $\approx$ 1 ppm,

in which case lower $I_{185}:I_{254}$ should be used. Future work will investigate the sensitivitiy of $NO_x$-dependent, OH-initiated
OVOC and SOA formation processes to $I_{185}:I_{254}$.

*Code and data availability.*  Data and KinSim mechanisms presented in this paper are available upon request. The KinSim kineticsolver is
freely available at : https://tinyurl.com/kinsim-cases#bookmark=kix.6zu8zdwq2lce.

*Author contributions.*  AL conceived and planned the experiments. JR performed the experiments. JR and AL performed the data analysis.

JR, AL, and WB conceived and planned the model simulations, and JR and AL carried out the model simulations. JR, AL and WB contributed
to the interpretation of the results. AL took the lead in writing the paper. All authors provided feedback on the paper.





*Competing interests.* The authors declare no competing interests.

*Acknowledgements.* We thank Chris Rockett (Light Sources Inc.), Dan Spicer (Light Sources Inc.), Leah Williams (Aerodyne), and John Jayne (Aerodyne) for helpful discussions.



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





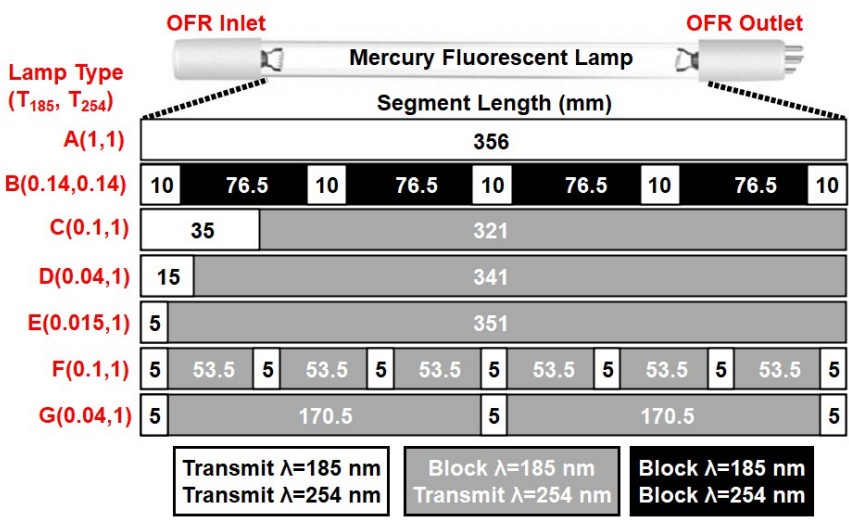

**Figure 1.** Low-pressure Hg fluorescent lamp types used in this study. Each lamp type contains 356 mm of quartz material that either transmits both $\lambda = 185$ and 254 nm radiation (white, $T_{185} = 1$), blocks $\lambda = 185$ nm and transmits $\lambda = 254$ nm radiation (grey, $T_{185} = 0$), or blocks both $\lambda = 185$ and 254 nm radiation (black, $T_{185} = 0$).





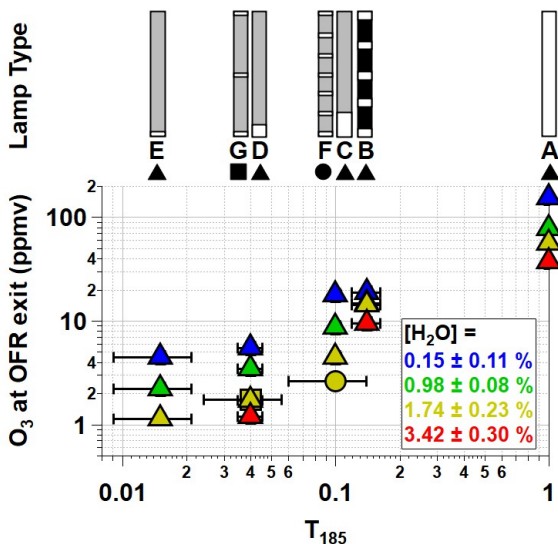

**Figure 2.** $O_3$ mixing ratio generated using OFR185 at $I_{254} = (3.5\pm0.7)\times10^{15}$ photons $cm^{-2}$ $s^{-1}$ (lamp types A and C-G) and $I_{254} = 5.8\times10^{14}$ photons $cm^{-2}$ $s^{-1}$ (lamp type B) as a function of $T_{185}$ and $[H_2O]$. Error bars represent $\pm1\sigma$ of replicate $O_3$ measurements and $\pm$ 2 mm uncertainty in lengths of individual $T_{185}$ = 0 and 1 segments.

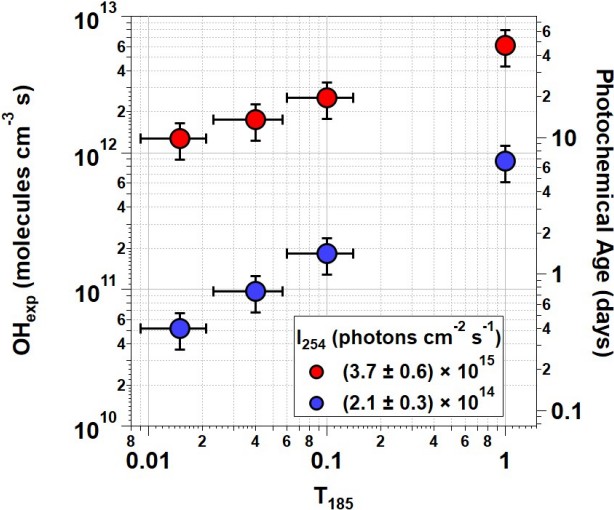

**Figure 3.** $OH_{exp}$ generated using OFR185 ($[H_2O]$ = 1.90$\pm$0.26%) at minimum and maximum $I_{254}$ for each $T_{185}$ value. Corresponding photochemical age shown on right y-axis assuming mean [OH] = $1.5\times10^6$ molec $cm^{-3}$ (Mao et al., 2009). Error bars assume $\pm$ 30% uncertainty in $OH_{exp}$ and $\pm$ 2 mm uncertainty in lengths of individual $T_{185}$ = 0 and 1 segments.



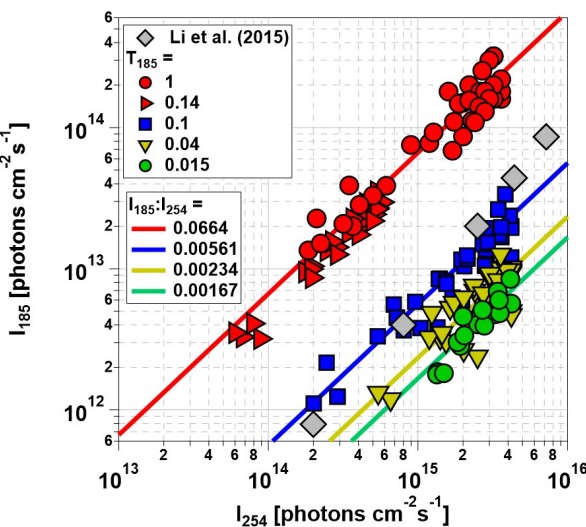

**Figure 4.** Calculated $I_{185}$ and $I_{254}$ values for the lamp types shown in Figure 1. $I_{185}:I_{254}$ values were calculated from linear regression functions and used to derive $OH_{ex}$ estimation equations. $I_{185}$ and $I_{254}$ values obtained by Li et al. (2015a) in an earlier-generation PAM OFR are shown for reference.

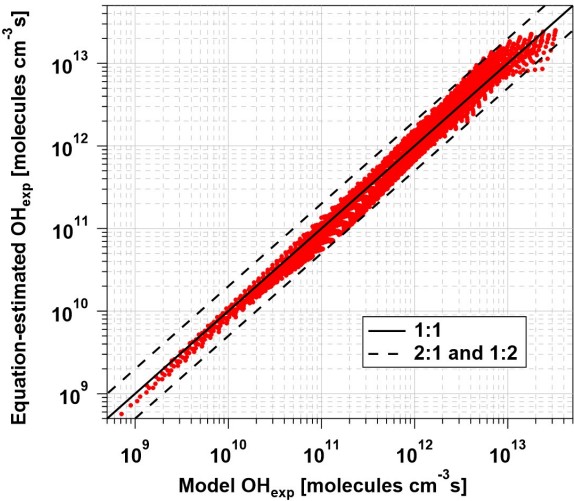

**Figure 5.** $OH_{exp}$ calculated from the estimation equation (Eq. 1) as a function of $OH_{exp}$ calculated from the full OFR185 KinSim mechanism (Table S1) for lamp types A and B. Solid and dashed lines correspond to the 1:1 and the 1:2 and 2:1 lines, respectively. Estimation equation fit coefficients are shown in Table 2.





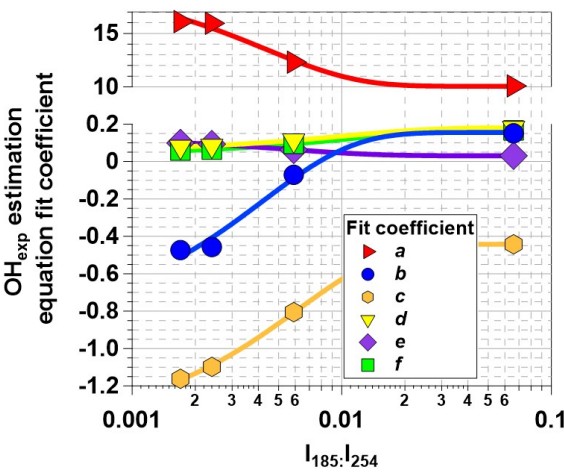

**Figure 6.** $OH_{exp}$ estimation equation fit coefficients plotted as a function $I_{254}:I_{254}$. Trendlines were calculated from exponential regression functions with fit parameters that are presented in Table 3.

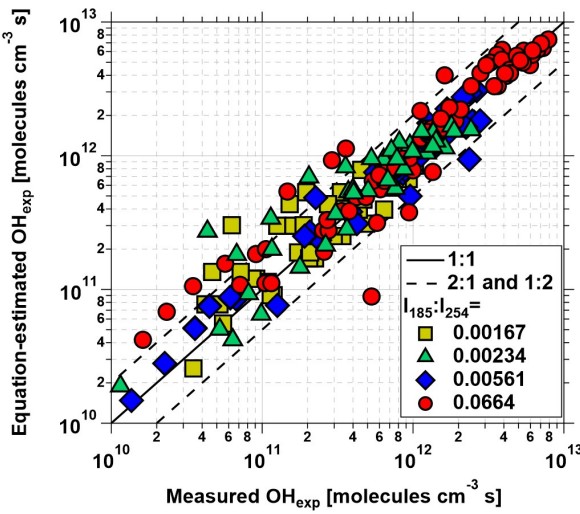

**Figure 7.** $OH_{exp}$ calculated from estimation equation (Eq. 1 and Table 2) as a function of $OH_{exp}$ calculated from tracer decay method for Hg lamp types with $I_{254}:I_{254}$ values specified in legend.





**Table 1.** OFR conditions input to photochemical model.

| P (mbar) | 1013 |
|---|---|
| T ($^\circ$C) | 22.5-31.9 |
| Residence time (s) | 63, 124, 251 |
| $H_2O$ (%) | 0.03-3.9 |
| $O_3$ (ppmv) | 0.4-156 |
| CO (ppmv) | 0 or 6-9 |
| $SO_2$ (ppbv) | 0 or 288-629 |
| $I_{185}$ (photons cm$^{-2}$ s$^{-1}$) | $1.1\times10^{12}$-$3.2\times10^{14}$ |
| $I_{254}$ (photons cm$^{-2}$ s$^{-1}$) | $6.0\times10^{13}$-$4.2\times10^{15}$ |

**Table 2.** $OH_{exp}$ estimation equation coefficients ($\pm1\sigma$) as defined in Eq. 1).

| $I_{185}$:$I_{254}$ | Coefficient | | | | | |
|---|---|---|---|---|---|---|
| | a | b | c | d | e | f |
| 0.00167 | $16.109 \pm 0.321$ | $-0.4734 \pm 0.0382$ | $-1.1613 \pm 0.0182$ | $0.079284 \pm 0.00105$ | $0.99503 \pm 0.00195$ | $0.059251 \pm 0.00115$ |
| 0.00242 | $15.949 \pm 0.347$ | $-0.45692 \pm 0.0398$ | $-1.0974 \pm 0.0186$ | $0.084855 \pm 0.0012$ | $0.093976 \pm 0.00206$ | $0.064116 \pm 0.00134$ |
| 0.00595 | $12.306 \pm 0.42$ | $-0.070275 \pm 0.04130$ | $-0.8052 \pm 0.0227$ | $0.11347 \pm 0.00249$ | $0.062916 \pm 0.00233$ | $0.094896 \pm 0.00291$ |
| 0.0664 | $10.098 \pm 0.576$ | $0.15062 \pm 0.0455$ | $-0.44244 \pm 0.0329$ | $0.18041 \pm 0.00872$ | $0.031146 \pm 0.00265$ | $0.1672 \pm 0.00953$ |

**Table 3.** Parameterization of Eq. 1 coefficients ($\pm1\sigma$): y0 + A $\times$ exp[$I_{185}:I_{254}$) $\times$ invTau].

| Coefficient | y0 | A | invTau |
|---|---|---|---|
| a | $10.053 \pm 0.593$ | $9.4455 \pm 1.52$ | $230.41 \pm 71.8$ |
| b | $0.15553 \pm 0.0641$ | $-0.99468 \pm 0.168$ | $237.54 \pm 76.9$ |
| c | $0.44174 \pm 0.0106$ | $-0.95747 \pm 0.0223$ | $163.04 \pm 8.41$ |
| d | $0.18069 \pm 0.000904$ | $-0.12054 \pm 0.00161$ | $98.577 \pm 3.91$ |
| e | $0.031037 \pm 0.00208$ | $0.094968 \pm 0.00462$ | $182.31 \pm 18.8$ |
| f | $0.16754 \pm 0.00167$ | $-0.1287 \pm 0.00295$ | $96.245 \pm 6.65$ |