# Peer review of "Technical Note: Effect of varying the $\lambda = 185$ and 254 nm photon flux ratio on radical generation in oxidation flow reactors"

_Atmospheric Chemistry and Physics, 2020_

## Referee Comment (RC1) · Anonymous Referee #1 · 16 Aug 2020

This study focuses on the investigation of the behavior of different mercury lamps and specifically on how much OH radical they can generate either via photolysis of ozone or photolysis of oxygen and water based on the emitted wavelength (254 and/or 185 nm). The different lamps are compared for different concentration of O3, water, external OH reactivity and current applied. The comparison leads to the development of an equation to estimate the amount of OH radical produced as function of water, O3 and external OH reactivity applicable for the majority of lamps currently in use.

The paper is well written and well-structured and fits the purpose of a Technical Note within ACP. I recommend its publication.

[Figure]

Minor comments.

Page 2, line 35. I find this sentence a bit confusing. Specifically the statement that compressed air or O2 is not required for external O3 generation making the lamps with 185 nm easier to use in field studies. How are normally lamps with 185 nm used in the field? Just with ambient air? How a stable water vapor concentration is achieved?

Page 2, line 49. Please add the specification of the sensor used to measure the relative humidity.

Page 3, section 2.1. Here it is not clear how the choice of these lamps was made. I assume it was done to cover the largest possible range of I185:I254 ratios? Or are these lamps the most commonly used? I would suggest to extend a little bit the section to more clearly explain these lamps were used.

Page 7, code and availability. There it is stated that the KinSim mechanism is available upon request. Though, in the supplement of the paper (page 4) a table is listed with caption indicating that it contains the KinSim mechanism. So, which one was used within this study? I would recommend including the one developed within the study in the supplement so that anyone could make use of it.

---

## Referee Comment (RC2) · Anonymous Referee #2 · 21 Sep 2020

This paper reports the effect of varying the ratio of UV-light with the wavelength of 185nm versus 254 nm in an oxidation flow reactor. The effects on chemistry is presented and the data is used to parametrize the OH exposure for the set-up. The experiments appear to be sound and done in a thorough way. For a technical note I also fine the interpretations sufficient. One may clarify that the methods generated will most likely only be valid for this specific set-up of the OFR. However, since the aerodyne OFR is commercially available and applied by many groups there is a general interest of the performance.

My major concerns/points to be addressed are:

[Figure]

*The motivation behind using the estimation equation where six parameters are fitted. I assume the physical meaning behind all these parameters (factors) are described in Li et al but one can expect a short introduction to the equation also in this technical note to understand what are instrumental specifics and what are related to a general parametrization of the chemistry or physics.

*Clarify to what extent the information provided are limited to the OFRs designed and commercialized by aerodyne. Note: If the extent is significant I would suggest a short statement under "competing interests" according to the ACP policy.

*The discussion on plug-flow condition. I assume one can give a more accurate description on the residence time distribution with measured data obtained from a pulse of an inert tracer compound.

Minor edits/points:

Row 3: Add NO3 -radicals

Row 57: I assume this dimming voltage is arbitrary? Can it be described in a better way?

Row 64: Specify the type of quartz.

Row 75/76: Strange wording. For me this would be a "reference OH reactivity" using reactivity of known tracers. Replace "external" with "reference" or "tracer".

Row 81: See major comment.

Row 96: Strange wording. If I understand right the model was tuned or adjusted?

Row 139: How much of the findings is Hg-lamp specific? Is this deviation to be expected when changing lamps in an OFR?

Row 165: Rephrase so it's clearer that HO2 and NOx in addition to OH increases.

Row 183: Provide the values of used cross-sections.

Row 202: the used KinSin mechanism is also available in the supplemental?

---

## Author Comment (AC1) · 29 Sep 2020

**Response to reviewers for the paper "Technical Note: Effect of varying the λ = 185 and 254 nm photon flux ratio on radical generation in oxidation flow reactors."**

We thank the referees for their comments on our paper. To guide the review process, we have copied the referee's comments in black text. Our responses are in blue text. We respond to Referee #1 and #2 comments, with alterations to the paper indicated in **bold or  text** below and in annotations to the revised manuscript.

**Anonymous Referee #1**

1. Page 2, line 35. I find this sentence a bit confusing. Specifically the statement that compressed air or O2 is not required for external O3 generation making the lamps with 185 nm easier to use in field studies. How are normally lamps with 185 nm used in the field? Just with ambient air? How a stable water vapor concentration is achieved?

We modified the text as follows:

P2, L35: "Additionally, OFR185 is often more practical than OFR254 to apply in field studies because **$O_2$ and $H_2O$ that are already present in ambient air are photolyzed to generate $O_3$, OH and $HO_2$, whereas OFR254 requires addition of** compressed air or $O_2$  for external $O_3$ generation **and additional inlet plumbing to inject it at the OFR inlet**."

2. Page2, line 49. Please add the specification of the sensor used to measure the relative humidity.

We modified the text as follows:

P2, L49: "A photodetector (TOCON-C6, sglux Gmbh) and a relative humidity and temperature (RH/T) sensor **(SHT21, Sensiron)** were mounted in the exit flange of the OFR."

3. Page 3, section 2.1. Here it is not clear how the choice of these lamps was made. I assume it was done to cover the largest possible range of I185:I254 ratios? Or are these lamps the most commonly used? I would suggest to extend a little bit the section to more clearly explain these lamps were used.

We modified the text as follows:

P3, L59: "Figure 1 shows the Hg fluorescent lamp configurations that were used in this study. Lamp type A is **an** ozone producing low-pressure Hg germicidal fluorescent lamp (GPH436T5VH/4P, Light Sources Inc.) in which quartz that transmits λ = 185 and 254 nm radiation is present along the entire 356 mm arc length. **This lamp type is a standard component of the Aerodyne PAM OFR.** The relative transmissivity of λ = 185 nm radiation (T185) in lamp type A is thus equal to 1. […] A different type of quartz is available that blocks λ = 185 nm and transmits λ =254 nm radiation ($T_{185}$=0). **To cover the largest possible range of $I_{185}:I_{254}$,** Lamp types C, D, E […] fused one segment each of quartz with T185 = 0 and T185 = 1 to provide reduced $I_{185}$ relative to lamp type A while maintaining constant $I_{254}$."

4. Page 7, code and availability. There it is stated that the KinSim mechanism is available upon request. Though, in the supplement of the paper (page 4) a table is listed with caption indicating that it contains the KinSim mechanism. So, which one was used within this study? I would

recommend including the one developed within the study in the supplement so that anyone could make use of it.

*We will upload the KinSim mechanism that was used in this study to the Supplement materials.*

**Anonymous Referee #2**

**My major concerns/points to be addressed are:**

1. The motivation behind using the estimation equation where six parameters are fitted. I assume the physical meaning behind all these parameters (factors) are described in Li et al but one can expect a short introduction to the equation also in this technical note to understand what are instrumental specifics and what are related to a general parametrization of the chemistry or physics.

*We modified the text as follows:*

*P5, L140: "Previous studies reported empirical OH exposure algebraic estimation equations for use with OFRs […]. We adapted the estimation equation format introduced by Li et al. (2015):*

$$\log[\mathrm{OH_{exp}}] = \left(a + (b + c \times \mathrm{OHR_{ext}^d} + e \times \log[\mathrm{O_3} \times \mathrm{OHR_{ext}^f}]) \times \log[\mathrm{O_3}] + \log[\mathrm{H_2O}]\right) + \log\left(\frac{\tau}{124}\right) \qquad (1)$$

**This equation incorporates the following relationships between $\mathrm{OH_{exp}}$ and $\mathrm{O_3}$, $\mathrm{H_2O}$, $\tau$ and $\mathrm{OHR_{ext}}$ identified by Li et al. (2015): (1) a power-law dependence of $\mathrm{OH_{exp}}$ on UV intensity, and, accordingly, $[\mathrm{O_3}]$; (2) a linear dependence of $\mathrm{OH_{exp}}$ on $[\mathrm{H_2O}]$ and $\tau$; (3) OH suppression as a function of increasing $\mathrm{OHR_{ext}}$. The fit coefficents *a-f* are lamp-specific."**

2. Clarify to what extent the information provided are limited to the OFRs designed and commercialized by aerodyne. Note: If the extent is significant I would suggest a short statement under "competing interests" according to the ACP policy.

*We modified the text as follows:*

*P7, L191: "To develop a general framework within which to evaluate and compare different OFR185 systems, we characterized $\mathrm{OH_{exp}}$ as a function of $\mathrm{I_{185}}$, $\mathrm{I_{254}}$, $\mathrm{OHR_{ext}}$, and $[\mathrm{H_2O}]$ values, in the process using several novel low-pressure Hg lamp configurations to extend the range of achievable $\mathrm{I_{185}}$:$\mathrm{I_{254}}$. $\mathrm{OH_{exp}}$ estimation equations were developed for the Hg lamp types that were used, and corresponding estimation equation fit coefficients were parameterized as a function of $\mathrm{I_{185}}$:$\mathrm{I_{254}}$ to enable interpolation to other OFR185 systems  **that can employ the same Hg lamp type(s) over the range of $[\mathrm{O_3}]$, $[\mathrm{H_2O}]$, $\mathrm{OHR_{ext}}$ and $\tau$ values parameterized here. Because low-pressure Hg germicidal fluorescent lamps are used in many industries (e.g. medical, HVAC, wastewater remediation), they are less expensive and more easily acquired than other Hg lamps."***

3. The discussion on plug-flow condition. I assume one can give a more accurate description on the residence time distribution with measured data obtained from a pulse of an inert tracer compound.

We agree with the reviewer that the residence time distribution can be measured to within better than ± 30% uncertainty at a specific OFR condition. However, in our opinion, it is unlikely that such a measurement could be then accurately applied across many OFR conditions where parameters such as temperature vary due to ambient conditions or waste heat dissipation from lamps. For example, in Lambe et al. (2019), mean residence times obtained from pulsed tracer measurements at OFR temperatures of 22°C and 39°C were 120 ± 34 s and 98 ± 63 s during periods when the lamps were turned off and on respectively. Rather than making explicit RTD measurements across many OFR conditions, simply accepting ± 30% uncertainty in the residence time may be an acceptable alternative for many users.

To clarify this point in the manuscript, we revised the text as follows:

P3, L80: "In most experiments, the **calculated** mean residence time was $\tau_{OFR}$ = 124 s, which was **obtained from the ratio of the internal OFR volume** ($\approx$ 13 L) and the total sample and makeup flow rate through the OFR (6.4 L min$^{-1}$). This calculation implicitly assumes plug flow conditions, with associated uncertainty of approximately 10% compared to an explicit residence time distribution measurement **at a specific OFR condition** (Li et al., 2015). **Variability in OFR parameters (e.g. temperature, flow rate) may increase the uncertainty in this assumption across a continuum of conditions (Huang et al., 2017; Lambe et al., 2019).** To characterize the uncertainty in **our** plug flow approximation **across multiple sample flow conditions**, we measured integrated OH exposure (OH$_{exp}$) values of $3.3\times10^{11}$, $7.8\times10^{11}$, and $2.0\times10^{12}$ molecule cm$^{-3}$ s at **sample flow rates of 12.5, 6.4 and 3.1 L min$^{-1}$**, respectively, using the tracer decay method (Sec.2.2) with the OFR operated at the same humidity and lamp intensity. Thus, perturbing the "plug flow" $\tau$OFR = 124 s by a factor of 2 in either direction changed OH$_{exp}$ by factors of 2.36 and 2.56. Based on these results, an upper-limit estimated uncertainty in $\tau_{OFR}$ and corresponding OH$_{exp}$ is approximately 30%."

The following citations will be added to References:

**Huang, Y., Coggon, M. M., Zhao, R., Lignell, H., Bauer, M. U., Flagan, R. C., and Seinfeld, J. H.: The Caltech Photooxidation Flow Tube reactor: design, fluid dynamics and characterization, Atmos. Meas. Tech., 10, 839–867, https://doi.org/10.5194/amt-10-839-2017, 2017.**

**Lambe, A. T., Krechmer, J. E., Peng, Z., Casar, J., Carrasquillo, A. J., Raff, J. D., Jimenez, J. L., and Worsnop, D. R.: HO$_x$ and NO$_x$ production in oxidation flow reactors via photolysis of isopropyl nitrite, isopropyl nitrite-d$_7$, and 1,3-propyl dinitrite at $\lambda$ = 254, 350, and 369 nm, _Atmos. Meas. Tech._., 12, 299-311, https://doi.org/10.5194/amt-12-299-2019, 2019.**

**Minor edits/points:**

  4. Row 3: Add NO3 -radicals

We modified the text as follows:

P1, L3: "Oxidation flow reactors (OFRs) complement environmental smog chambers as a portable, low-cost technique for exposing atmospheric compounds to oxidants such as ozone (O$_3$), **nitrate (NO$_3$) radicals,** and hydroxyl (OH) radicals. "

  5. Row 57: I assume this dimming voltage is arbitrary? Can it be described in a better way?

We modified the text as follows:

P3, L57: "The dimming voltage applied to the ballast ranged from 0.8 to 10 VDC. **Below ~0.8 VDC, the lamp output was unstable due to flickering, and 10 VDC was the maximum control voltage permitted by the ballast."**

    6.    Row 64: Specify the type of quartz.

We modified the text as follows:

P3, L59: "Figure 1 shows the Hg fluorescent lamp configurations that were used in this study. Lamp type A is a standard ozone producing low-pressure Hg germicidal fluorescent lamp […] in which **type 214** quartz that transmits λ = 185 and 254 nm radiation is present along the entire 356 mm arc length. […] A different type of quartz is available (**type 219**) that blocks λ = 185 nm and transmits λ =254 nm radiation ($T_{185}$ =0)."

    7.    Row 75/76: Strange wording. For me this would be a "reference OH reactivity" using reactivity of known tracers. Replace "external" with "reference" or "tracer".

We prefer the term "external OH reactivity" to distinguish from "internal OH reactivity" associated with intrinsic OFR photochemical reactions such as OH + $HO_2$ and OH + OH.

    8.    Row 81: See major comment.

For reference, here is text on lines 80-82 of the discussion paper: "In most experiments, the mean residence time was $\tau_{OFR}$ = 124 s, which was calculated from the ratio of the internal OFR volume (≈ 13 L) and the total sample and makeup flow rate through the OFR (6.4 L min $^{-1}$). This calculation implicitly assumes plug flow conditions, with associated uncertainty of approximately 10% compared to an explicit residence time distribution measurement (Li et al., 2015b)."

We assume the referee is referring to their Comment #3 above, where they stated: "I assume one can give a more accurate description on the residence time distribution with measured data obtained from a pulse of an inert tracer compound." If that is the case, we refer the referee to our response to comment #3 by referee #2. If that is not the case, we invite the referee to clarify and/or submit a follow up comment that we can reply to.

    9.    Row 96: Strange wording. If I understand right the model was tuned or adjusted?

We modified the text as follows:

P4, L95: "For cases where [$H_2O$] ≤ 0.1% and the RH **sensor accuracy became a limiting factor, we systematically adjusted the [$H_2O$] value that was** input to the model **to a value between 0.01 and 0.1%** to  **achieve** better   agreement **between measured and modeled $OH_{exp}$"**.

    Row 139: How much of the findings is Hg-lamp specific? Is this deviation to be expected when changing lamps in an OFR?

On P5, L139, we hypothesized that the different apparent lamp power dependence on $I_{185}$:$I_{254}$ between this study and Li et al. (2015) was "presumably due to differences in the specific Hg lamps and/or method of dimming used in the two studies." Thus, we agree with the referee's suggestion that the findings might be specific to the Hg lamp type, and we think the manuscript as written already acknowledges that possibility. Thus, we assume that the referee is hypothesizing that lamp-to-lamp variability within the

same lamp type could also contribute additional uncertainty. While that has not been systematically evaluated here, we modified the text as follows to account for that as a possible additional explanation:

P5, L137: "$I_{185}$: $I_{254}$ = 0.00561 for lamp types C and F (blue symbols) fell within the envelope of $I_{185}$: $I_{254}$ = 0.004 to 0.012 characterized by Li et al. (2015), with a lower apparent sensitivity of $I_{185}$:$I_{254}$ to lamp power. This is presumably due to differences in the specific Hg lamp **type**s, **potential variability in lamp output within the same lamp type,** and/or method of dimming used in the two studies**.**

   10. Row 165: Rephrase so it's clearer that HO2 and NOx in addition to OH increases.

We modified the text as follows:

P6, L165: " [HO$_2$] and [NO$_x$] (with N$_2$O present) increase **along with [OH] as a function of** $I_{185}$ (Fig. S4)."

   11. Row 183: Provide the values of used cross-sections.

We modified the text as follows:

P7, L184: " Assuming upper limit $\phi_{185}$ = 1 and $\phi_{254}$ =1 values **(Atkinson, 1986), $\sigma_{185}$ = 2.8\*10$^{-17}$ cm$^2$, and $\sigma_{254}$ = 8.9\*10$^{-19}$ cm$^2$ (Dawes et al., 2017)**, $F_{photolysis,benzene}$ ≤ 0.26 and 0.07 at low and high I185:I254".

   12. Row 202: the used KinSin mechanism is also available in the supplemental?

Please see our response to Comment #4 raised by Referee #1.